# Room-temperature super-elongation in high-entropy alloy nanopillars

Qian Zhang [1,6], Ranming Niu [2,6], Ying Liu [2], Jiaxi Jiang [1], Fan Xu [3], Xuan Zhang [1], Julie M. Cairney [2], Xianghai An [2]✉, Xiaozhou Liao [2]✉, Huajian Gao [4,5]✉ & Xiaoyan Li [1]✉

Nanoscale small-volume metallic materials typically exhibit high strengths but often suffer from a lack of tensile ductility due to undesirable premature failure. Here, we report unusual room-temperature uniform elongation up to ~110% at a high flow stress of 0.6–1.0 GPa in single-crystalline <110>-oriented CoCrFeNi high-entropy alloy nanopillars with well-defined geometries. By combining high-resolution microscopy and large-scale atomistic simulations, we reveal that this ultrahigh uniform tensile ductility is attributed to spatial and synergistic coordination of deformation twinning and dislocation slip, which effectively promote deformation delocalization and delay necking failure. These joint and/or sequential activations of the underlying displacive deformation mechanisms originate from chemical compositional heterogeneities at the atomic level and resulting wide variations in generalized stacking fault energy and associated dislocation activities. Our work provides mechanistic insights into superplastic deformations of multiple-principal element alloys at the nanoscale and opens routes for designing nanodevices with high mechanical reliability.

Rapid advances in manufacturing techniques have enabled the fabrication of small-volume materials at the nanoscale, including nanopillars and nanowires that are being widely used in micro/nanodevices with unprecedented functionality[1–6]. Because such small systems are inevitably subjected to external loading or even large deformations during service, their reliability, integrity, and optimum design are crucially dependent on the mechanical performance of the component nanomaterials[7]. Although nanoscale metallic materials typically exhibit high strengths[8], they are often susceptible to premature failure during uniform stretching, which is embodied by highly localized shear fracture or brittle fracture. Such undesirable brittleness and low ductility originate from the limited sources and interactions of dislocations in a nanoscale volume[8]. This deficiency invariably inhibits the sustainable activation of displacive deformation mechanisms through dislocation and/or twinning activities, leading to plastic instability and premature failure.

From a mechanistic perspective, collective shearing of atoms in displacive processes is activated within a narrow stress threshold window[8,9]. Once the displacive plasticity is initiated under tensile deformation, continual nucleation of new dislocations is energetically prohibited, essentially localizing deformations and then catalyzing premature failure of the nanostructures. Although considerable elongation could be enabled via single or limited twin boundary (TB) migration in some pure metallic nanowires[3–6], a critical concern is that the stress dropped sharply and instantly after yielding and then remained at a low level during subsequent deformation. In addition, it

[1]Centre for Advanced Mechanics and Materials, Applied Mechanics Laboratory, Department of Engineering Mechanics, Tsinghua University, Beijing 100084, China. [2]School of Aerospace, Mechanical, and Mechatronic Engineering, The University of Sydney, Sydney, NSW 2006, Australia. [3]Institute of Mechanics and Computational Engineering, Department of Aeronautics and Astronautics, Fudan University, Shanghai 200433, China. [4]School of Mechanical and Aerospace Engineering, College of Engineering, Nanyang Technological University, 70 Nanyang Drive, Singapore 639798, Singapore. [5]Institute of High Performance Computing, A*STAR, Singapore 138632, Singapore. [6]These authors contributed equally: Qian Zhang, Ranming Niu. ✉e-mail: xianghai.an@sydney.edu.au; xiaozhou.liao@sydney.edu.au; huajian.gao@ntu.edu.sg; xiaoyanlithu@tsinghua.edu.cn

was reported that surface diffusion mediated by dislocation plasticity could reduce the likelihood of plastic instability[10–13]. When surface diffusional mechanisms became active, excellent stretchability was achieved in some nanocrystals with small and uncontrollable characteristic sizes[11,12], which was partly attributed to the complex stress/strain state resulting from the irregular shapes. However, surface diffusion is sensitive to surface quality and can be easily suppressed by oxidation or deposition layers[11]. Therefore, an open question is whether large tensile ductility can be achieved at all at the high-stress levels typically seen in metallic nanopillars/nanowires.

Here, we micro-fabricate single-crystalline <110>-oriented CoCrFeNi high-entropy alloy (HEA) nanopillars with well-defined geometries, demonstrating exceptional uniform super-elongation of more than ~100% with ultrahigh flow stress based on in situ transmission electron microscopy (TEM) tensile testing. We unveil the underlying deformation mechanism and mechanistic origins of the ultrahigh uniform tensile ductility through in situ TEM, scanning TEM (STEM) atomic-resolution high-angle annular dark-field (HAADF) imaging, and large-scale atomistic simulations. As well documented, multi-constituent species of HEA alloys could offer enormous possibilities for statistical fluctuation in compositional and packing arrangements of the various elements, which create a wide spectrum of "local" stacking fault energies (SFEs) to affect the physical mechanisms and mechanical properties fundamentally[14,15]. Therefore, unlike the large deformability mediated by surface diffusional creep in nanocrystals[10–12], such room-temperature super-elongation at high-stress levels arises from spatial and enduring activation of substantial displacive plasticity involving nucleation, propagation, and coalescence of nanotwins and extensive twin-dislocation interactions. The distinct deformation mechanisms are attributed to ingrained compositional heterogeneities at the atomic level of the HEAs, enabling a spread-out distribution of the stress threshold for synergistic activation of a series of displacive deformations, which provide more energy

dissipation channels, effectively eliminating strain localization and ensuring ultrahigh uniform plasticity. In addition, our findings provide robust proof and atomistic visualization to verify how chemical heterogeneities regulate the deformation mechanisms and influence mechanical performance, highlighting the inherent plasticity of HEA alloys.

## Results and discussion

The single-phase face-centered cubic (FCC) structure of the well-annealed CoCrFeNi HEA was identified with high-energy synchrotron X-ray diffraction (HE-XRD) experiments (Fig. 1a). We carried out electron backscatter diffraction (EBSD) to characterize the microstructure of the bulk HEA, which consists of fully recrystallized, equiaxed grains with an average size of ~120 μm and random crystallographic orientations (Fig. 1b). Energy dispersive X-ray spectroscopy (EDS) maps (Supplementary Fig. 1) indicate that all individual elements are uniformly distributed and nearly equiatomic. We employed focused ion beam (FIB) milling to fabricate dog-bone-shaped single-crystalline HEA nanopillars from selected grains with specific orientations (see details in Methods). A representative <110>-oriented HEA nanopillar with a well-defined orientation and gauging area of ~350 nm in length by ~70 nm in width is exhibited in Fig. 1c, d. Although a rather high density of FIB-induced defects was observed, the uniform contrast of the TEM image suggested a homogeneous thickness (Fig. 1d), which was measured to be ~73 nm along the edge-on direction and confirmed by using electron energy loss spectroscopy (Supplementary Fig. 2). The inset selected area electron diffraction (SAED) pattern verifies that nano-mechanical testing of the nanopillars was along the <110> direction.

We conducted in situ uniaxial tensile experiments on single-crystalline CoCrFeNi HEA nanopillars along <100>, <110>, and <111> directions at room temperature and a strain rate of ~2.8 × 10⁻³ s⁻¹. Figure 1e shows typical engineering tensile stress-strain curves for the <110>-oriented nanopillars, which exhibited an ultrahigh uniform

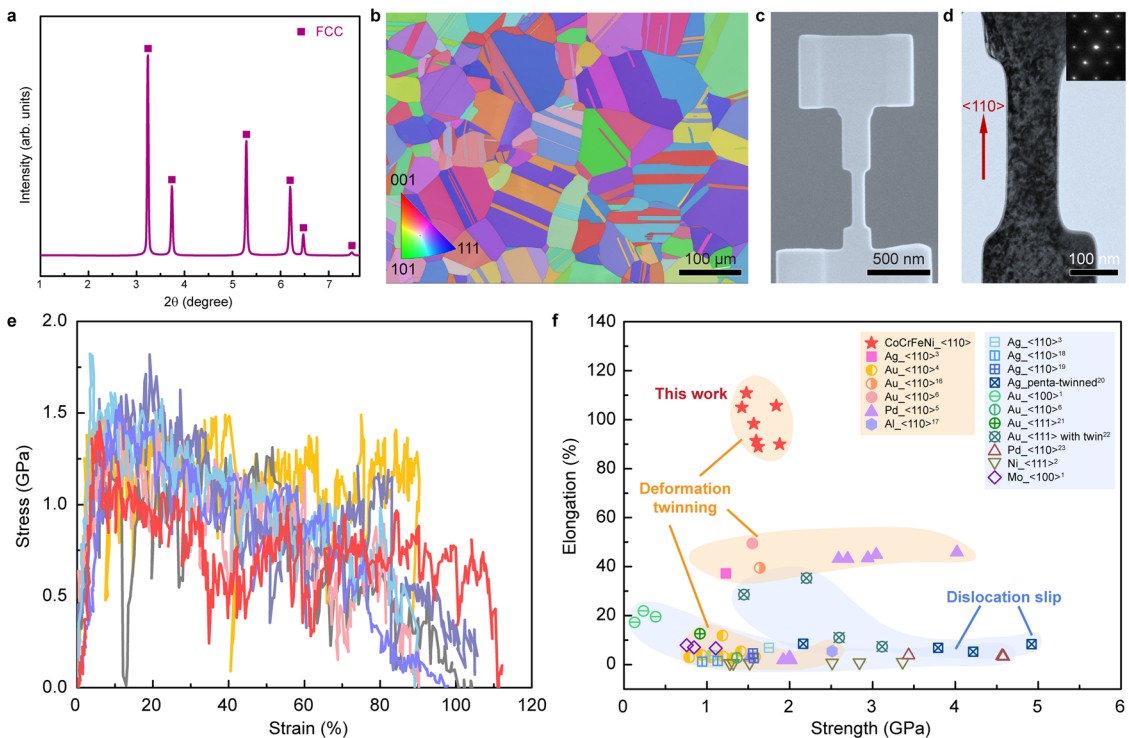

**Fig. 1 | Microstructural characterization of bulk CoCrFeNi alloys and tensile properties of <110>-oriented HEA nanopillars. a** XRD pattern of CoCrFeNi alloy. **b** An EBSD image of the well-annealed bulk CoCrFeNi alloy. **c** An SEM image of a typical <110>-oriented HEA nanopillar microfabricated via FIB milling. **d** TEM image

of a well-defined gauging area from a single-crystalline <110>-oriented HEA nanopillar. **e** Tensile stress-strain curves of <110>-oriented HEA nanopillars. **f** Ashby chart of tensile elongation versus yield strength for <110>-oriented HEA nanopillars and various metallic nanowires[1–6,16–23].

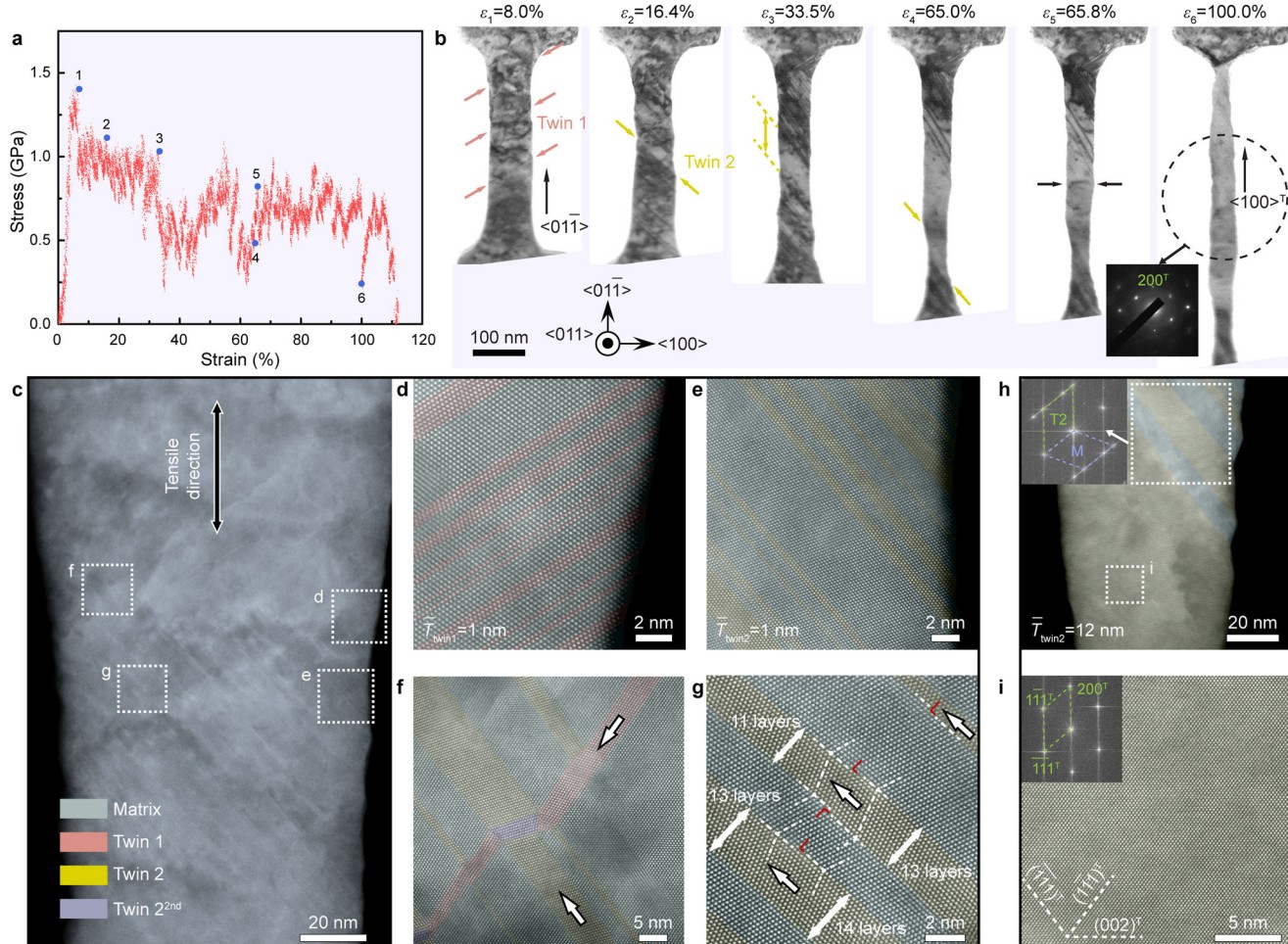

**Fig. 2 | In situ TEM tensile and post-mortem HAADF STEM results for a typical [$0\bar{1}\bar{1}$]-oriented HEA nanopillar. a** Tensile engineering stress-strain curve for an HEA sample showing ~110% elongation to failure. **b** Snapshots are taken from Supplementary Movie 1 at multiple strain levels, as annotated in **a**, demonstrating typical deformation processes of the HEA sample. Plastic deformation was initiated by deformation twinning Twin 1 on the ($1\bar{1}\bar{1}$) planes (red arrow) and subsequently dominated by spatial nucleation and thickening of Twin 2 on the ($\bar{1}\bar{1}1$) plane (yellow arrow). At 100.0% strain, the sample was entirely transformed and reoriented to $<100>^{T}$. **c–i** HAADF STEM images of another sample that deformed and stopped deliberately at a strain of 55.3%, enabling the typical spatial heterogeneous deformation features. The area marked by broken squares in **c** was enlarged in **d–g** at atomic resolution, demonstrating **d**, **e** nucleation of Twin 1 and Twin 2, respectively; **f** interaction of Twin 1 & 2 and **g** thickening of Twin 2 via partial dislocation slip along TBs. **h** microstructures of the sample beneath the area in **c**, showing coarse Twin 2 verified by embedded fast Fourier transformation and fully twinned lattice. **i** lattice image of the fully twinned area, as marked in **h**.

tensile ductility of more than ~100% under high flow stress of 0.6–1.0 GPa. This unusual room-temperature super-elongation in HEA nanopillars is highly orientation-dependent. The <100>- and <111>-oriented HEA nanopillars presented very limited plasticity due to the ready occurrence of large shear localization through dislocation glide and deformation twinning, respectively (Supplementary Figs. 3 and 4). We plotted an elongation-versus-strength diagram for our <110>-oriented HEA nanopillars and various metallic nanowires/nanopillars[1–6,16–23] under tensile deformation. The exceptional uniform elongation of the HEA nanopillars with ultrahigh yield strengths outperformed that of all other existing nanopillars/nanowires, most of which suffered from low ductility. Regardless of whether the onset of yielding in these nanostructures was enabled by dislocation activities or deformation twinning, premature failure occurred, especially in pristine nanowires. When plasticity mainly occurs by TB migration, large elongations of up to ~50% can be obtained in some <110>-oriented pure metallic nanowires[3,5,6,16]. This indicates that the room-temperature super-elongation of the HEA nanopillars should originate mechanistically from their distinct deformation mechanisms.

To unravel the underlying mechanisms for the exceptional uniform plasticity, we explored the deformation dynamic processes of the

<110>-oriented HEA nanopillar in detail. By capturing snapshots from Supplementary Movie 1, which correspond to the marked strain values in the stress-strain curve (Fig. 2a), a sequence of TEM images presented in Fig. 2b suggests that the characteristic fingerprint of microstructural evolution was derived from profuse twinning. At the initial stage of deformation, several nanoscale twins nucleated from the free surface and quickly propagated across the sample at different sites, as indicated by the arrows ($\varepsilon_1 = 8.0\%$ in Fig. 2b). With increasing strain, multiple nanotwins formed on another {111} plane due to activation of a new twinning system. It is compelling that early nucleated twins seem to be "erased" after interaction with newly formed twins[24] and only partly survived in the top area (Supplementary Movie 1 and $\varepsilon_2 = 16.4\%$ in Fig. 2b). After that, nucleation, propagation, and merging of nanoscale twins at the new twinning system essentially coordinated the ensuing deformation, which led to continuous expansion of the twinned area ($\varepsilon_3 = 33.5\%$ in Fig. 2b). Subsequently, although a narrow region at the bottom of the nanopillar, which is indicated by the arrows ($\varepsilon_4 = 65.0\%$ in Fig. 2b), formed due to coalescence of several twins, strain localization did not occur. Instead, the plasticity was mainly mediated by the dislocation slip in the twinned areas marked by arrows and similar twinning at the top of the sample ($\varepsilon_5 = 65.8\%$ in Fig. 2b).

Even though two sets of nanotwins interacted with each other, thickening and merging of the twins proceeded until the whole pillar was fully twinned into the <100> orientation, as confirmed by the SAED pattern, which rendered uniform tensile super-elongation ($\varepsilon_6 = 100.0\%$ in Fig. 2b). The detailed process for deformation of the nanopillar can be found in Supplementary Movie 1. Owing to spatial and subsequent activation of the sophisticated twinning behavior, the flow stress of the HEA nanopillar was maintained at a high level of 0.6–1.0 GPa (Fig. 2a).

We used high-resolution HAADF STEM to further investigate the deformation-induced microstructural characteristics at the atomic scale (Fig. 2c–i). We stretched one <110>-oriented HEA nanopillar to a tensile strain of ~55% and then deliberately unloaded it for a post-mortem study. Supplementary Fig. 5 shows the stress-strain curve and the low-magnification TEM images of this deformed nanopillar. Figure 2c presents the overall microstructure in the area where nucleation, interaction, and thickening of nanotwins were observed. The HAADF lattice images in Fig. 2d, e indicate activation of the two twinning systems (Twins 1 and 2, respectively), which enabled the formation of many nanoscale twins with an average thickness of ~1 nm. The two sets of nanotwins interacted with each other, and the thickness of Twin 1 was gradually reduced by the interactions with Twin 2, signaling the occurrence of detwinning (Fig. 2f). As mentioned above, Twin 1 was finally annihilated by the complex interaction with Twin 2 systems which dominated the ensuing plasticity. In particular, twin propagation and coalescence played crucial roles in sustaining uniform deformation. Figure 2g shows that many partial dislocations emitted from the intersections between TBs and the free surface and glided along the TBs, which resulted in the thickening of the nanotwins. The HAADF image in Fig. 2h presents the microstructures of an area with thickened twins and a massive area with a twinned lattice. The atomic structure of the fully twinned region substantiates that the initial <110> orientation along the loading axis was converted to the <100> axis (Fig. 2i). Therefore, although single or a few migrating TBs can prevail in noble metal nanowires[3–6], our in situ testing and atomic-scale characterization showed that extensive twinning activities (nucleation, interaction, propagation, and coalescence of nanotwins) spatio-temporally orchestrated plastic deformation to achieve ultrahigh uniform tensile ductility.

We performed a series of large-scale molecular dynamics (MD) simulations to determine the mechanistic origins of the unusual room-temperature super-elongation exhibited by the single-crystalline <110>-oriented HEA nanopillars. The atomic configuration of a representative simulated CoCrFeNi nanopillar with a random distribution of individual element atoms and a rectangular cross-section with different width-thickness ratios (defined as $\lambda$ varying from 1 to 2) is illustrated in the insert of Fig. 3a (see details in "Methods"). Considering that there were initial dislocations in the nanopillars microfabricated via FIB, we also explored the effects of pre-existing dislocations (classified by different initial dislocation densities $\rho_0$) on the mechanical behavior. Figure 3a shows stress-strain curves from the MD simulations with different $\lambda$ and $\rho_0$. Evolutions of twin fraction and dislocation density with the tensile strain of the simulated samples are presented in Fig. 3b, c, respectively. Irrespective of the values of $\lambda$ and $\rho_0$, the twin fraction increased prominently with strain at a similar steady rate, leading to a high twin fraction up to approximately 90%. The high density of dislocations also played a critical role in maintaining the plasticity of each sample, especially when the twin fraction was saturated. Nevertheless, the dislocation density was generally higher in the pristine nanopillars. A sequence of snapshots for the simulated sample with $\lambda = 2$ and $\rho_0 > 0$ showed typical microstructural evolution nearly identical to that observed in experiments (see Fig. 3d and further details in Supplementary Movie 2). The distribution of the corresponding atomic shear strain across the whole sample demonstrated that there was no strong shear localization during uniform plastic flow. The incipient plasticity was initiated by multiple nucleation of partial dislocations from the free surface, while the successive partial dislocation glide on the {111} planes and the interactions between dislocations and bundles of stacking faults (SF) enabled the formation of nanotwins[25] (Supplementary Fig. 6). With straining, two sets of twins (Twins 1 and 2) interacted with each other, and the partial dislocations either transmitted across the TB of Twin 2 or cross-slipped onto the TB via dynamic interactions between dislocations and TBs[26]. As demonstrated in Fig. 3e, two distinct processes led to the detwinning and the final elimination of Twin 1. The enduring nucleation of partial dislocations from the surface and subsequent slipping along the TB of Twin 2 enabled prominent thickening of Twin 2 and reduced the length of Twin 1. However, partial dislocations nucleated from the intersection sites glided along the TB of Twin 1, which led to the shrinkage of Twin 1 in its thickness direction. These dynamic twinning and detwinning processes suppressed localization of the strain and allowed continuous uniform tensile deformation. It is noted that after yielding, all the simulated samples exhibit a similar trend in the evolution of twin fraction (Fig. 3b), but the distinct trend in the evolution of dislocation density (Fig. 3c). These results indicate that both initial dislocation density and width-thickness ratio have a significant influence on the evolution of dislocation density, but have a weak influence on the evolution of twin fraction. In the future, we will further quantify the influences of initial dislocation density, width-thickness ratio, and other configuration parameters (including sample shape/morphology and atomic ordering) on the evolutions of dislocation density and twin fraction, which is crucial for understanding the plastic deformation behaviors and mechanisms of HEAs.

Consistent with experimental observations, propagation and merging of nanotwins dominated the ensuing deformation and noticeably carried the plastic strain (Fig. 3d). Even though a small notch-like flaw formed on the surface due to the twin interactions highlighted with a yellow arrow ($\varepsilon = 35.0\%$ in Fig. 3d), further plasticity was still mainly governed by twin extension, which alleviated localized deformation and prevented premature necking. Apart from surface-partial dislocation-nucleation-controlled TB migration, the frequent occurrence of TB coalescence facilitated and accelerated the expansion of the twinned area. Figure 3f presents a typical process for atomic-scale merging of nanotwins. Two twin domains, with lattice orientations shown in the inset, propagated for a certain distance and gradually approached each other ($\varepsilon = 50.00\%$ in Fig. 3f). Then, partial dislocations on each TB interacted with each other and generated extrinsic SFs or intrinsic SFs ($\varepsilon = 60.40\%$ in Fig. 3f). Gliding of these partial dislocations toward the free surface ($\varepsilon = 62.80\%$ in Fig. 3f) promoted merging of the two twin domains into a whole twinned area with few dislocations ($\varepsilon = 64.00\%$ in Fig. 3f). Thorough integration of multiple nanotwins resulted in reorientation of the lattice from <110> to <100> for almost the entire sample. Finally, shear localization arising from dislocation slip was initiated at the bottom of the fully twinned sample and caused the final failure. Our large-scale MD simulations corroborated the spatial activation of profuse twinning in suppressing and eliminating potential strain localization, which endowed the single-crystalline <110>-oriented HEA nanopillars with uniform plasticity. Similar deformation processes can be found for the simulated sample without initial dislocations (Supplementary Figs. 7a–d), implying that such superplastic behavior of the HEA nanopillars could be conceivably related to the microstructures and intrinsic properties of the HEA.

For comparison, we also conducted atomistic simulations of the uniaxial tension in single-crystalline pure Al, Ni, and Ag <110>-oriented nanopillars. Al and Ni are the constitutive elements of $Al_x$CoCrFeNi and have higher SFEs than CoCrFeNi, while Ag has a low SFE close to that of CoCrFeNi. The stress-strain curves shown in Supplementary Fig. 7 manifest a similar stress trend but less elongation for pure metals (approximately 30% for Al and Ni and 60% for Ag) in comparison to the HEA, with deformation localization at the sites of twin–twin

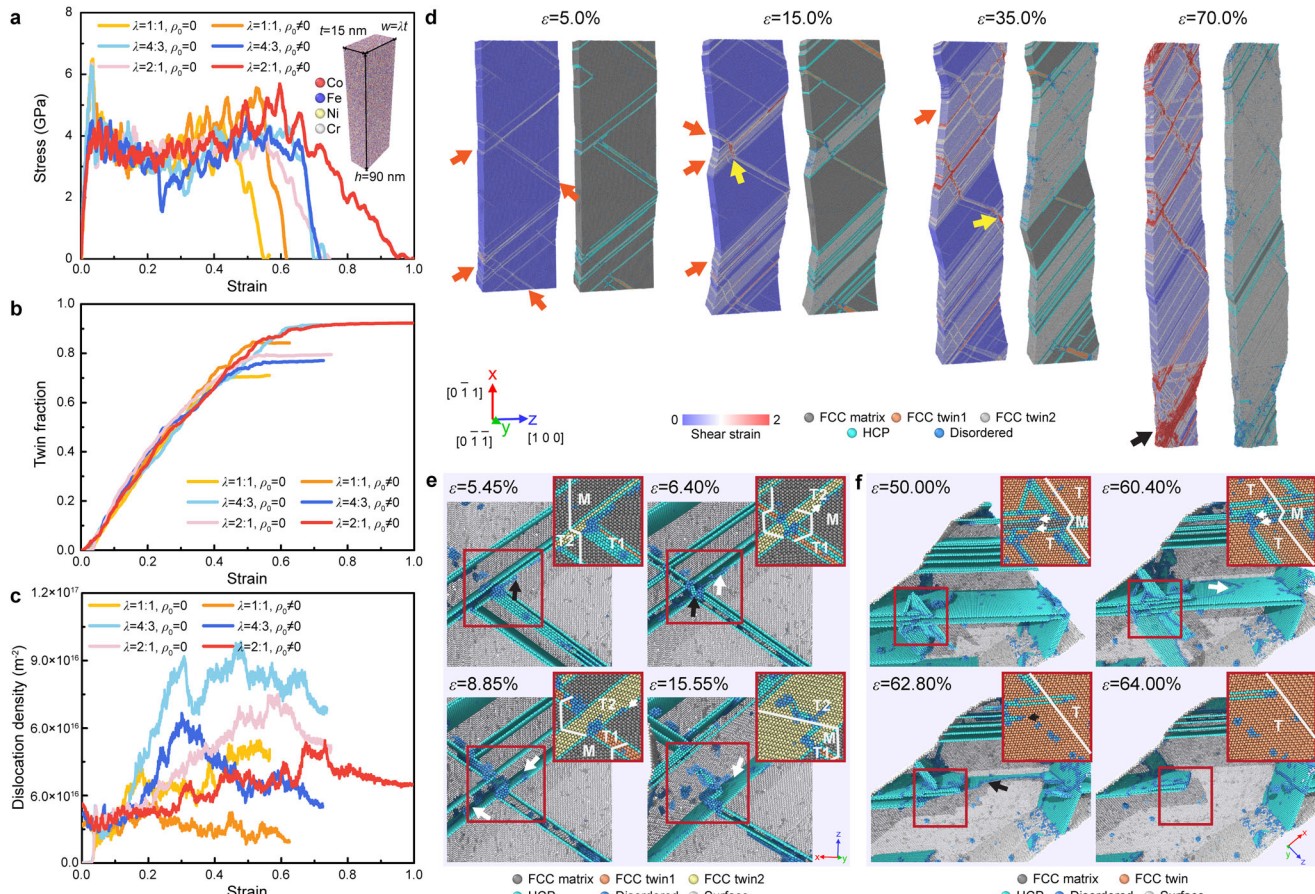

**Fig. 3 | Atomistic simulations of <110>-oriented HEA nanopillars under uniaxial tension. a** Tensile stress-strain curves of simulated nanopillars with different width-thickness ratios and initial dislocation densities. The inset exhibits a typical atomic configuration of a simulated nanopillar with a rectangular cross-section. **b, c** Evolutions of the twin fraction and dislocation density with tensile strain in the simulated nanopillars with different $\lambda$ and $\rho_0$ values, respectively. **d** A sequence of snapshots of a stretched nanopillar from Supplementary Movie 2 indicate that deformation twinning dominates plastic deformation, leading to deformation delocalization and large uniform elongation at room temperature. The atoms are colored according to their von Mises shear strain and local crystalline structure separately. The orange arrows and black arrows indicate the deformation twinning and dislocation slip, respectively. The yellow arrows indicate the twin−twin interaction and a resultant notch-like flaw on the surface. **e** Twin−twin interaction processes inducing detwinning of Twin 1. **f** Merging processes of two deformation twins. The atoms are colored according to their local crystalline structure in **e** and **f**. The black and white arrows in **e** and **f** indicate the movement of SFs and the partial slip-inducing TB migration, respectively.

interactions marked by yellow arrows. Since dislocation slip and twinning are the major plasticity carriers, evolutions of the dislocation density and twin fraction with the tensile strain of the simulated samples could indicate differences in their plasticity. The dislocation densities of the HEA nanopillar were much higher than those of pure metallic samples, suggesting a higher dislocation multiplication rate for the HEA. Nevertheless, the nanotwins nucleated at a relatively larger strain in these pure metallic nanopillars, but their amounts increased sharply at the initial stage of plastic deformation. After that, twin propagation was severely hindered by twin−twin and twin-dislocation interactions, and then dislocation slip dominated the deformation and finally gave rise to premature plastic instability. These comparison results implied that the distinct deformation mechanisms responsible for room-temperature super-elongation originated from the intrinsic nature of the HEA due to compositional heterogeneities at the atomic level[27–29]. These results also suggested that the activation of these unique deformation mechanisms is associated with the introduction of multiple elements.

For FCC crystals, the SFE determines which deformation mechanism dominates plastic deformation[30]. Unlike the constant SFE value found for conventional metals and alloys, we found that the SFE value for the CoCrFeNi HEA is spatially variable by calculating generalized planar fault energy (GPFE) curves. Figure 4a illustrates typical

simulation configurations during nucleation and growth of twins, which were generated by rigidly shifting {111} planes in the sequence indicated by the arrows. We constructed 5000 independent samples with different random solid solutions and computed their GPFE curves. Figure 4b shows the statistical average of 5000 GPFE curves for HEA samples, indicating a wide distribution of energy pathways for the deformation of CoCrFeNi plotted in blue shadow. The GPFE curve for the pure Ni sample is included for comparison in Fig. 4b, which indicates fixed energy for plastic deformation of pure Ni. Figure 4c, d presents the calculated SFE and twin fault energy of the HEA sample, which play crucial roles in nucleation of twinning partials and twin growth. Both material parameters exhibited substantial scattering, which intrinsically originated from statistical fluctuations in the compositional and packing arrangements of different elemental atoms in the HEA lattice. A wide spectrum of "local" SFEs allowed synergistic and spatial activation of nucleation and propagation for nanotwins to delocalize plastic strain and hence enable extraordinary ductility (>100%) under high strength. Recent experimental studies showed that the heterogeneous atomic configurations of HEAs promoted chemical short-range ordering (SRO), which had a remarkable influence on mechanical properties and deformation behaviors[31–33]. We also performed simulations to explore the effects of SRO on the superplastic behavior of <110>-oriented HEA nanopillars (Supplementary

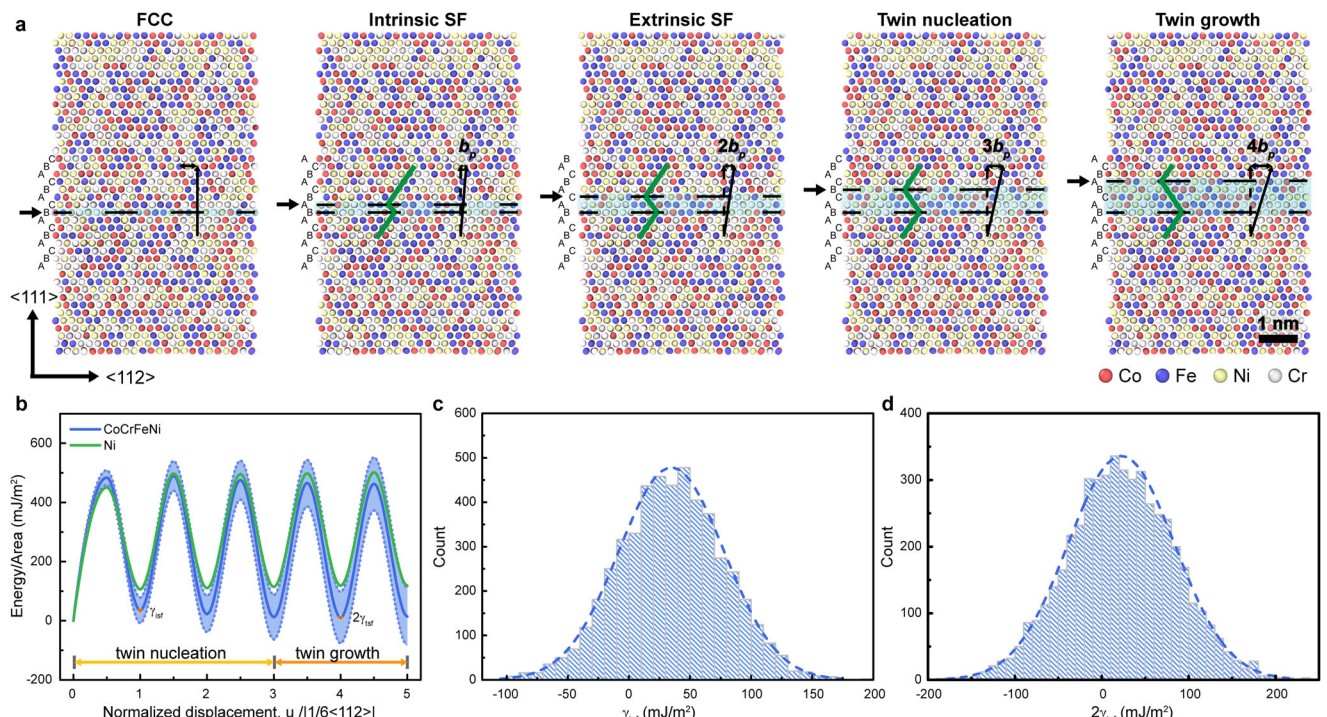

**Fig. 4 | Calculation of GPFE curves for HEA and Ni. a** Schematic illustrations of GPFE calculation of the HEA sample. Nucleation and growth of twins are induced by successive slip of partial dislocations on the {111} planes, as indicated by arrows. **b** GPFE curves of the HEA and Ni samples. The curve of the HEA sample is taken as the average from the GPFE curves of 5000 samples with different random solid solutions. The blue shadow shows a deviation of such a statistical average. **c, d** Statistical distributions of intrinsic SFE and twice the twin energy of HEA, marked in **b**, from 5000 calculations.

Fig. 8). We found that SRO did not fundamentally alter the activation of plastic mechanisms and deformation processes but slightly increased the yield strength.

In addition to twinning activities and GPFE curves, we further investigated the dislocation morphology and mobility at a constant strain rate or shear stress, with results indicating that there was a higher barrier for achieving equivalent dislocation velocity in HEAs than in Ni (Supplementary Fig. 9). The extended dislocation in HEA exhibited rugged dislocation lines and a significantly varied dissociation width, which differed markedly from that in Ni with smooth straight dislocation lines and a uniform SF width. Dislocation glide in the HEA was enabled via a series of forward slips by local segments after depinning from local heterogeneities[34,35], and this resulted in nontrivial fluctuations in shear stress under a constant strain rate or in dislocation velocity under constant shear stress. The distinct dislocation behavior of the HEA is essentially pertinent to the severe lattice distortion due to the different atomic sizes and intrinsic properties of individual constituent elements[15,36]. Therefore, higher flow stress is required to overcome the considerable resistance to dislocation glide and to enable plastic deformation.

Our in situ TEM results and atomistic simulations confirmed that the ultrahigh uniform plasticity under high stress was predominantly controlled by the formation and propagation of nanotwins in a continuous manner, both of which were spatially and synergistically mediated by partial dislocations. Based on our MD simulations and the energetics of the twinning processes[37–39], we derived the critical shear stress for partial dislocation nucleation from the free surface, which initiated deformation twinning (Supplementary Fig. 10). The derived expression is given by (see the details in Supplementary Note 1):

$$\tau_{crit} = \frac{Gb_p}{2R} \frac{1}{4\pi(1-\nu)} \left(1 + \ln\frac{mR}{b_p}\right)\left(2 - \nu - \frac{2\nu}{\pi}\right) + \frac{\gamma_{sf}}{b_p} \quad (1)$$

where $\tau_{crit}$ is the critical resolved shear stress for nucleation of twinning partials, $G$ is the shear modulus, $b_p$ is the magnitude of Burgers vector of partial dislocation, $R$ is the dislocation loop radius, $\nu$ is Poisson's ratio, $m$ is the line energy correction factor varying with wedge angles for corner nucleation, and $\gamma_{sf}$ is the SFE. Using SFE values of approximately 30–34.8 mJ/m² (from experimental measurement[40] and our calculations) and setting the radius of the surface dislocation loop as 5–10 nm (which is comparable to the potential surface source size[39]), we obtained overall critical twinning stress values of approximately 628 - 903 MPa. According to the Schmid factor of 0.471 for twinning under <110>-oriented tension, the applied normal stress was calculated to range from 1.33 to 1.92 GPa, which is consistent with the yield strength range in our experiments. The presence of initial dislocations inside pillars can lower the critical stress owing to a larger radius of surface dislocations or potential dislocation sliding. During deformation, the energy barrier for partial nucleation was further reduced due to the concentration of the local stress at TB-surface junctions[41]. More significantly, spatially variable local SFEs of the HEA provided a wide range of stress thresholds for persistent nucleation of partial dislocations, thus stimulating strain delocalization and suppressing plastic instability. It is noted that nearly complete twinning via a single or a few TB migrations only contributed to a plastic strain of ~41.4%[5]. Therefore, uniform tensile super-elongation (exceeding 100%) of our HEA nanopillars is attributed to the spatial and synergistic coordination of enhanced dislocation activities and deformation twinning, as indicated by our atomistic simulations (Fig. 3 and Supplementary Figs. 7 and 8). Furthermore, we performed the same atomistic simulations for uniaxial tension of <110>-oriented Al₀.₁CoCrFeNi and CoCrNi nanopillars and observed the same uniform super-elongation and deformation mechanisms in these nanopillars (Supplementary Figs. 11, 12, and 13). Together with a comparison with Ni nanopillars, these results indicate that such super-elongation is general in the multi-principal element alloy nanopillars with lower and

variable SFEs. It simultaneously implies that such super-elongation originates from chemical compositional heterogeneities at the atomic level, which is an intrinsic nature in the multi-principal element alloys.

In summary, nanoscale small-volume materials typically exhibit high strength but low tensile ductility. The latter is particularly important for the structural reliability of many micro/nanodevices. Here, we have achieved unusual room-temperature tensile ductility in excess of 100% under a high tensile flow stress in single-crystalline <110>-oriented CoCrFeNi HEA nanopillars. Our experimental, computational, and theoretical investigations showed that the ultrahigh uniform tensile ductility was caused by the intensive activation of distinct twinning mechanisms (nucleation, interaction, propagation, and coalescence of nanotwins) and extensive twin-dislocation interactions. These unique plasticity mechanisms originate from the atomic-level chemical complexity of the HEA that generates variable local SFEs and further synergistically coordinates a spatial activation of a series of deformation mechanisms. Our findings provide mechanistic insights for understanding plastic deformations of small-volume metallic materials and create a mechanistic map enabling the design of micro/nanodevices with reliable mechanical and functional properties. Due to the vast compositional space of HEAs, our findings imply that multiple constituent species offer enormous possibilities for tuning compositional heterogeneities at the atomic scale and provide a fundamental and comprehensive understanding of distinct deformation mechanisms and unique material properties.

## Methods

### Preparation and characterization of the bulk alloy sample
Bulk CoCrFeNi samples were fabricated by arc melting a mixture of pure metal powders with a high purity of >99.9% and equiatomic concentrations in an argon atmosphere. The alloy ingots were remelted five times to ensure chemical homogeneity and cast into a copper mold with dimensions of $16 \times 20 \times 80$ mm$^3$. After that, the alloy ingots were thermally homogenized at 1473 K for 24 h and then cold rolled to achieve a 65% reduction in thickness and further annealed at 1373 K for 1 h to obtain equiaxed grain structures. The single FCC phase of the bulk alloy was identified by using synchrotron X-ray experiments with high-energy synchrotron X-ray radiation at beamline 11-1D-C. An EBSD analysis of the annealed samples was applied to characterize their microstructures and crystallographic orientation using a Carl Zeiss Ultra Plus field emission gun scanning electron microscopy (SEM) with an operating voltage of 15 kV and a step size of 0.25 μm. The chemistry information of the samples was collected using an Oxford Instruments Aztec EDS system with an X-Max 20 mm$^2$ silicon drift detector. Before the microstructural characterization, annealed bulk samples were mechanically ground and polished sequentially by using silicon carbide sandpapers, diamond suspensions, and an oxide suspension with a silica particle size of 50 nm. Following that, the samples were further electro-polished to a stress-free and mirror-like surface using a Struers LectroPol-5 unit and a 5% perchloric acid in acetic acid under a voltage of 8–10 V at the ambient temperature.

### In situ TEM tensile testing and characterization
To prepare nano-dog-bone-shaped specimens for the quantitative in situ TEM tensile experiments, a well-polished sample was transferred to a Thermofisher® G4 Hydra Plasma FIB (PFIB)-SEM equipped with an Oxford Instruments® Symmetry CMOS EBSD detector. Based on the default sample coordination system defined in the AZtec® software, the grains with specific crystallographic orientations of <100>, <110> and <111> along the z-axis, i.e., the loading axes during the following in situ nanomechanical testing, were first identified using EBSD, while only those grains with another <110> orientation along y-axis that is the zone axis in TEM were finally selected to facilitate characterization of the defects behavior during in situ TEM tensile tests. After selecting the grains with specific orientations, a

$70 \times 10 \times 4$ μm$^3$ lamella was vertically lifted out using the standard trenching method, taking advantage of the high milling efficiency and stabilized nano-manipulator of the PFIB. The lifted-out lamella was welded onto an Omniprobe® grid using Pt deposition, which has been adhered to a special copper mount that was specially designed for the Hysitron PI 95 in situ holder. After the lift-out processes, the sample was transferred to a Zeiss® Auriga Ga$^+$ FIB-SEM to prepare the in situ tensile sample, taking advantage of the high precision of Ga® FIB. In situ, nano-dog-bone samples with a gauging area of ~$70 \times 70 \times 350$ nm$^3$ (i.e., width × thickness × length) were prepared using the ion beam of 30 kV 300 pA and 50 pA. To ensure homogeneous thickness in the gauging area of the samples, an inclined angle of ~±1.9° was applied to make tape-free samples during the milling. Following that, a cleaning process was conducted to minimize the ion damage and potential redeposition using an ion beam of 5 kV 100 pA with a ±6° inclination angle.

After sample preparation, images of each sample were captured in SEM and TEM to measure the geometry of the gauging area for the calculation of stress and strain values. The length and width were measured by imaging the dog-bone samples along the surface-on direction, while the thickness was measured along the edge-on direction. The homogeneity of thickness along the gauge was confirmed by using the Electron Energy Loss Spectroscopy (EELS) technique to map the thickness (Supplementary Fig. 2). EELS thickness mapping was conducted in a JEOL® 2200FS field TEM with an in-column Omega energy filter operating at 200 kV. The entrance angle was 170 mrad and the collection angle was 38 mrad. The inelastic mean-free path (MFP) for the equiatomic CrCoNiFe was 100.5 nm, which was calculated by averaging the inelastic MFP values of Cr (104 nm), Co (98 nm), Ni (98 nm), and Fe (102 nm) under an electron beam at 200 kV. The thickness of the prepared samples was homogeneous with an average value of ~73 nm, in agreement with the SEM measurements (Supplementary Fig. 2). Therefore, SEM measurement results were used for calculating the stress and strain values for the sake of efficiency.

In situ tensile tests were conducted in a JEOL® 2100 TEM operated at 200 kV, using a Hysitron® PI 95 Picoindenter equipped with a homemade nano-griper. The tensile test was conducted under displacement control mode using a constant tensile displacement rate of 1 nm/s, which is equivalent to a strain rate of $2.8 \times 10^{-3}$ s$^{-1}$ for a gauge length of 350 nm. The tensile axis of the nanosamples is within 3° from the designed crystallographic orientations, and the zone axis is near <110>, which is ideal for the observation of dislocation activities and twinning behaviors of FCC metals. Both displacement and load readings were recorded together with a video that shows the deformation process of the sample. The displacement was calibrated based on the video recordings. The original cross-sectional area (width×thickness) and initial length of the gauge part of the samples were employed to calculate engineering stress and strain, respectively.

To characterize the post-mortem microstructures and deformation mechanisms, one sample was deformed to a strain of ~55%, which allowed the activation of typical deformation events that are responsible for the super-elongation. After in situ TEM tensile tests, the low-magnification TEM image and selection-area diffraction patterns were taken in a JEOL® 2200FS field TEM operated at 200 kV, while the atomic-resolution high-angle annular dark-field (HAADF) images were taken using an aberration-corrected Themis-Z scanning TEM operated at 300 kV with collection angle in the range of 38-200 mrad.

### Atomistic simulations
To understand the underlying atomistic mechanisms of the experimentally observed deformation behaviors, we performed a series of large-scale MD simulations for uniaxial tension of HEA nanopillars via large-scale atomic/molecular massively parallel simulator (LAMMPS)[42]. We constructed simulation samples of single-crystalline HEA blocks with dimensions of $90 \times 31 \times 31$ nm$^3$. These blocks have three types of

[100], [110] and [111] orientations along their length direction. To investigate the effect of initial dislocations on mechanical properties and deformation behaviors, we introduced initial dislocations in the simulation samples via pre-straining. We first equilibrated the HEA blocks by dynamic relaxation for 200 ps and then compressed them to a strain of 5% at 300 K. After compression, a number of dislocations nucleated and glided in the blocks. Subsequently, we unloaded the compressed samples to zero stress and then equilibrated them for 100 ps at 300 K. Thus, some dislocations were maintained in the blocks. Then we cut the HEA blocks into nanopillars with different cross-sections. The simulated nanopillars had a length of 90 nm, thickness of 15 nm, and varied width of $15\lambda$ nm, where $\lambda$ in the range of 1 to 2 was used to control the cross-section of samples. All simulated nanopillars were equilibrated by energy minimization followed by dynamic relaxation for 200 ps at 300 K via an NPT ensemble. During relaxation, periodic boundary condition was imposed along the axial direction. The axial stress was controlled to be zero via a Parrinello-Rahmann barostat. After equilibration, all the simulated samples were stretched along their axial directions to failure with a constant strain rate of $5 \times 10^8$ s$^{-1}$. During stretching, all the simulated pillars were maintained at a constant temperature of 300 K via an NVT ensemble. The stress of the simulated sample was calculated by averaging the Virial tensile stress of all atoms. The common neighbor analysis (CNA)[43] was employed to identify the local crystalline structure of atoms and defects during plastic deformation. Atoms with hexagonal-closed-packed (HCP) order were painted in cyan, while the disordered atoms in dislocation cores or near-point defects were in blue. The atoms with FCC order are classified as the matrix, Twin 1, and Twin 2 according to their deformation history and lattice orientation. The dislocation analysis (DXA)[44] was applied to characterize the dislocation types and the Burgers vectors of dislocations. The von Mises atomic shear strain was obtained by calculating the local transformation matrix between two atomic configurations[45,46].

During simulations, we used a combined interatomic potential to describe the interatomic interactions of the CoCrFeNi and Al$_{0.1}$CoCrFeNi HEAs. This combined potential includes two embedded atom method (EAM) potentials[47,48] and two Lennard-Jones potentials[49]. The ternary EAM potential[47] is used to describe the self- and cross-interactions among Fe, Ni, and Cr. The cross-interactions of Al-Cr and Co-Cr are described by 12-6 Lennard-Jones potentials[49]. The quaternary EAM potential[48] is adopted to describe the self-interactions of Al-Al, Co-Co, and the remaining cross-interactions of Al-Co, Al-Fe, Al-Ni, Co-Fe, and Co-Ni. The microstructures and mechanical properties of Al$_x$CoCrFeNi HEAs have been predicted by using such combined interatomic potential. The predictions are in good agreement with those from the first-principles calculations and experiments[49]. An EAM potential[34] was used in the simulations of NiCoCr medium entropy alloy (MEA), and can well predict the microstructure and mechanical properties of NiCoCr MEA[34]. The EAM potentials for pure Al[50], Ni[50], and Ag[51] were used in the simulations of pure metallic single crystals.

## GPFE calculation

The GPFE curves of an HEA sample were calculated by using the combined interatomic potentials mentioned above. The constructed sample contains 8064 atoms and has dimensions of $5.2 \times 8.3 \times 1.9$ nm$^3$. The boundary condition along <111> direction was set to be free while periodic boundary condition was imposed in the other two directions. The nucleation and migration of nanotwins were generated by rigidly shifting the stacking close-packed (111) atomic planes in sequence[34,52]. Energy pathways were obtained by constrained energy minimization on intermediated configurations (only allowed to relax along the <111> direction) linearly interpolated between initial and final configurations[34]. The average GPFE curve was obtained by taking the average over all the curves from 5000 independent HEA samples with different random solid solutions.

## Dislocation mobility simulation

A cubic sample with dimensions of 20×20×20 nm$^3$ was employed to investigate the dislocation mobility in HEA, with edge directions aligned along <110>, <111>, and <112>. A 1/2<110>{111} edge dislocation was introduced at the center of the simulation box[35,53] by removing <110> half atomic plane and displacing other atoms accordingly. The periodic boundary conditions were imposed in the dislocation line direction and gliding direction. Eight layers of atoms on the top and bottom of the sample were fixed. Langevin thermostats were used in the regions with a thickness of 2.4 nm, which are neighboring the fixed regions on the top and bottom of the sample. The dislocation core located near the center of the sample has a large distance of 6 nm away from two thermostat regions[53]. Energy minimization and dynamic relaxation for 100 ps were conducted to obtain fully relaxed configurations. A constant shear loading or velocity was then applied on the fixed layers on the top and bottom of the sample to investigate the dislocation behaviors at 300 K under the constant shear stress or strain rate, respectively.

## Introduction of short-range order

The SRO parameter[34] is defined as $\alpha_{ij} = (p_{ij} - C_j)/(\delta_{ij} - C_j)$, where $p_{ij}$ is the average probability of finding a $j$-type atom in the first nearest-neighbor shell of an $i$-type atom, $C_j$ is the average concentration of $j$-type atom in the HEA system, and $\delta_{ij}$ is the Kronecker delta function. The deviation from zero of $\alpha_{ij}$ reflects the tendency of segregation or clustering[34]. The sample with SRO was achieved through an annealing process. During annealing, a single-crystalline sample with a random solid solution was equilibrated at a temperature of 2300 K for 300 ps, and then quenched to 0.1 K for 200 ps and relaxed at 0.1 K for 100 ps, followed by energy minimization.

## Data availability

The authors declare that the relevant data supporting the findings of this study are available within the article and its Supplementary Information files. All raw data generated in this study are available from the corresponding author upon request.

## Code availability

The open-source software and simulation setup were introduced in the Methods section. The input files for MD simulations supporting the findings of this study are available from the corresponding author upon request.

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

## Acknowledgements

X.Y.L. acknowledges financial support from the National Natural Science Foundation of China (grant numbers 12325203, 91963117, and 11921002). X.A. acknowledges support from the Australian Research Council under DE170100053 and DP23010228, from The University of Sydney under the Robinson Fellowship Scheme and from The University of Sydney Nano Institute under the Kickstarter Funding. X.Z.L. acknowledges financial support from the Australian Research Council grant DP190102243. R.N. is grateful for the financial support from the China Scholarship Council (student ID 201506840096). The authors are

grateful for the scientific and technical support from the Australian Centre for Microscopy and Microanalysis (ACMM) as well as the Microscopy Australia node at the University of Sydney. All the simulations were performed on the TianHe-1 supercomputer at the National Supercomputer Center in Tianjin.

## Author contributions

X.A., H.G., and X.Y.L. conceived the research. X.Y.L. and H.G. directed the atomistic simulation and Q.Z. performed the simulation. X.A. and X.Y.L. designed the experiments and R.N., Y.L., X.A., and Q.Z. conducted the experiments. Q.Z., R.N., and J.J. conducted the data analysis. F.X., X.Z., J.M.C., and X.Z.L. contributed to the discussion of results. X.A., Q.Z., H.G., and X.Y.L. wrote the manuscript. All authors participated in the discussion and revision of the manuscript.

## Competing interests

The authors declare no competing interests.
