## [Peer Review File · Nature Communications]

Room-temperature super-elongation in high-entropy alloy nanopillarsREVIEWER COMMENTS

Reviewer #1 (Remarks to the Author):

I have read the paper by Zhang et al. The authors report exceptionally high ductility in high-entropy alloy (HEA) nanopillars at room temperature. To this end, they perform in-situ tensile tests inside a TEM on CoCrFeNi HEA nanopillars and support their experimental observations with large-scale atomistic simulations. The authors observe elongation of the nanopillars beyond 100% and attribute it to deformation twinning and twin-dislocation interactions which serves to delay strain localization. These results are further supported by MD simulations which reveal that plasticity is initiated with nucleation of twins on multiple slip systems. Through atomistic calculations of stacking fault energies (SFE) and theoretical calculations, the authors demonstrate that this mechanical behavior originates from the compositional heterogeneity of the HEA which leads to variable local but low SFE. This enables sustained nucleation of partial dislocations throughout the nanopillar thereby suppressing strain localization.

The paper is well-written and is of great significance to the materials science field and to the readership of Nature Communications, given the considerable attention drawn by HEAs as next-generation structural materials. The authors report extraordinary mechanical properties, specifically room temperature ductility, of HEA nanopillars which make them promising candidates for structural applications. Their experimental and computational results provide valuable insights into the distinct deformation mechanisms at play in HEA due to their chemical complexity. The study is very extensive and thorough. The results are compelling and very well-supported by experiments, computations, and theoretical calculations.

I have one suggestion that may help strengthen the paper further. It would be valuable to discuss the novelty of the present work in the context of some of the existing literature on HEAs. It is reported in prior works that HEAs demonstrate good ductility (Y.F. Ye, High-entropy alloy: challenges and prospects, *Materials Today*, 2016) and that they possess low SFE which can vary locally with composition (E.P. George et al, High-entropy alloys, *Nature Review Materials*, 2019). The authors present a thorough comparison of HEA nanopillars with existing work on conventional single-component metallic nanowires/nanopillars in the introduction. A similar discussion with regards to existing studies on HEAs will highlight the novelty of their results and distill the contributions of the present study within the context of prior work on HEAs.

Reviewer #2 (Remarks to the Author):

This paper utilizes nanoscale experimental testing inside a TEM complemented with large-scale MD simulations to conduct room temperature mechanical tensile testing of high entropy alloy nanopillars. They demonstrate that this material geometry could exhibit ultrahigh uniform tensile ductility through deformation twinning and dislocation slip. This is a noteworthy finding, and can significantly impact the

design and fabrication of microelectronic devices. To the reviewer's knowledge, this work is new and is a significant advancement of prior mechanistic understanding of both high entropy alloys as well as nanostructured and nanoscale metals. Both the computational and experimental aspects are fairly complete, and conclusive.

Some (optional) minor suggestions are as follows:

(1) There appears to be very good matching between the experimental observations and the MD simulation results. Can the authors confirm if changes in the atomistic configurations (from that in Fig. 3) will influence the dislocation density and twin fractions? Perhaps these can be expressed as statistical quantities with error bars (akin to Fig. 4)

(2) One also does wonder how strongly the activated mechanisms depend on the alloying content. Perhaps a small set of additional simulations on regular metal alloys could be performed to demonstrate more strikingly that these mechanisms are only activated because of the presence of these multiple elements.

(3) The manuscript is very well written. Couple of typos found under the methods section on "atomistic simulations" (reference source not found, and also "EAM potentia" should be "EAM potential").

Responses to two reviewers

We are very grateful to both reviewers for their time and effort to review our paper and for their insightful and helpful comments/suggestions. In the light of the comments and suggestions from two reviewers, we have carefully revised the manuscript by performing additional simulations, by adding more statements and discussions about the influences of atomic configurations and introduction of multiple elements on evolution in defects and deformation behaviors/mechanisms, by adding some statements to highlight the contributions and novelty of the current study, and by fixing several typos. All the changes have been highlighted in red in the revised manuscript. The point-by-point responses to two reviewers are listed below.

Responses to Reviewer #1:

Comment #1. I have read the paper by Zhang et al. The authors report exceptionally high ductility in high-entropy alloy (HEA) nanopillars at room temperature. To this end, they perform in-situ tensile tests inside a TEM on CoCrFeNi HEA nanopillars and support their experimental observations with large-scale atomistic simulations. The authors observe elongation of the nanopillars beyond 100% and attribute it to deformation twinning and twin-dislocation interactions which serves to delay strain localization. These results are further supported by MD simulations which reveal that plasticity is initiated with nucleation of twins on multiple slip systems. Through atomistic calculations of stacking fault energies (SFE) and theoretical calculations, the authors demonstrate that this mechanical behavior originates from the compositional heterogeneity of the HEA which leads to variable local but low SFE. This enables sustained nucleation of partial dislocations throughout the nanopillar thereby suppressing strain localization.

The paper is well-written and is of great significance to the materials science field and to the readership of Nature Communications, given the considerable attention drawn by HEAs as next-generation structural materials. The authors report extraordinary mechanical properties, specifically room temperature ductility, of HEA nanopillars which make them promising candidates for structural applications. Their experimental

and computational results provide valuable insights into the distinct deformation mechanisms at play in HEA due to their chemical complexity. The study is very extensive and thorough. The results are compelling and very well-supported by experiments, computations, and theoretical calculations.

Response: We are very grateful to you for your time and effort to review our paper and for your positive comments on our manuscript.

Comment #2. I have one suggestion that may help strengthen the paper further. It would be valuable to discuss the novelty of the present work in the context of some of the existing literature on HEAs. It is reported in prior works that HEAs demonstrate good ductility (Y.F. Ye, High-entropy alloy: challenges and prospects, *Materials Today*, 2016) and that they possess low SFE which can vary locally with composition (E.P. George et al, High-entropy alloys, *Nature Review Materials*, 2019). The authors present a thorough comparison of HEA nanopillars with existing work on conventional single-component metallic nanowires/nanopillars in the introduction. A similar discussion with regards to existing studies on HEAs will highlight the novelty of their results and distill the contributions of the present study within the context of prior work on HEAs.

Response: Thank you very much for your insightful and helpful suggestions and for your providing two important references. As you mentioned, previous studies (*Science*, 2014, 345, 1153-1158; *Mater. Today*, 2016, 19, 349-362; *Nature*, 2016, 534, 227-230) have demonstrated that some bulk HEAs with the low stacking fault energy (SFE) have exhibited good tensile ductility at room temperature or cryogenic temperatures due to twinning-mediated plasticity or even phase transformation induced plasticity. It is well-known that for small-volume materials, their deformation mechanisms and mechanical properties are related to the sample sizes (*Prog. Mater. Sci.*, 2011, 56, 654-724; *Acta Mater.*, 2022, 223, 117460). Generally, nanoscale metallic materials possess high strength but poor ductility, originating from the limited sources and interactions of dislocations in the nanoscale volume (*Prog. Mater. Sci.*, 2010, 55, 710-757). To the best of our knowledge, there has been no experimental investigation of tensile behaviors of HEA nanowires/nanopillars. Therefore, we mainly presented the tensile deformation

behaviors of pure metallic nanowires in the introduction in our original manuscript and further compared the mechanical properties of our HEA nanopillars with those of various metallic nanowires in Fig. 1f. Compared with previous studies about metallic nanowires, our current work showed that the single-crystalline $\langle 110 \rangle$ -oriented CoCrFeNi HEA nanopillars with well-defined geometry exhibited exceptional uniform super-elongation of more than $\sim 110\%$ with ultrahigh flow stress at room temperature due to the spatiotemporal activations of substantial displacive plasticity. These results not only fill the gap in understanding the highly intertwined intrinsic/extrinsic deformation mechanisms and their relationships with extraordinary properties of nanosized HEAs, but also are compelling for manifesting the possibility of large tensile ductility in nanopillars and for offering new mechanistic strategies and insights to design novel small-volume alloys.

As you mentioned, multiple principal elements of HEAs could lead to spatial distribution of local chemical heterogeneities and promote statistical fluctuation of material properties, which further affect the deformation behaviors and mechanical properties of HEAs (Nat. Rev. Mater., 2019, 4, 515-534; Nature, 2019, 574, 223-227). Recent theoretical studies (Extreme Mech. Lett., 2017, 11, 84-88; Proc. Natl. Acad. Sci. U.S.A., 2018, 115, 8919-8924; Nat. Commun., 2019, 10, 3563) showed that atomic arrangements as well as short range order degree could profoundly alter local SFE distribution, while atomistic simulations (Extreme Mech. Lett., 2017, 17, 38-42; Nat. Commun., 2019, 10, 3563) and experimental studies (Nature, 2020, 581, 283-287; Nature, 2021, 592, 712-716; Nat. Commun., 2022, 13, 4777) revealed that the dislocation morphology and motion manner could be significantly influenced by local chemical heterogeneities. However, robust experimental evidence to verify how chemical heterogeneities regulate the deformation mechanisms and influence mechanical performance is still limited. Our study results provide compelling experimental proof and atomistic visualization to demonstrate the significance of local chemical heterogeneities in the spatiotemporal activations of substantial displacive plasticity to stimulate strain delocalization and suppress plastic instability, rendering the HEA nanopillars with the room-temperature uniform super-elongation.

In the light of your suggestions, we have added the following statements on HEAs on page 4 in the revised manuscript to distill the contributions of the current study and to highlight the novelty of the current study results, and cited the references you mentioned (Mater. Today, 2016, 19, 349-362; Nat. Rev. Mater., 2019, 4, 515-534) as Refs. 14 and 15 in the revised manuscript,

“High entropy alloys (HEAs) with multi-principal elements have exhibited excellent macro-mechanical properties exceeding those of conventional alloys^{14,15}. More intriguingly, multiple constituent species of HEA alloys offer enormous possibilities for statistical fluctuation in compositional and packing arrangements of the various elements, which create a wide spectrum of “local” stacking fault energies (SFEs) to affect the physical mechanisms and mechanical properties fundamentally^{14,15}. Herein, we micro-fabricated single-crystalline <110>-oriented CoCrFeNi HEA nanopillars with well-defined geometries, demonstrating exceptional uniform super-elongation of more than ~100% with ultrahigh flow stress based on in-situ transmission electron microscopy (TEM) tensile testing.”

“The distinct deformation mechanisms are attributed to ingrained compositional heterogeneities at the atomic level of the HEAs, enabling a spread-out distribution of the stress threshold for synergistic activation of a series of displacive deformations, which provide more energy dissipation channels, effectively eliminate strain localization and ensure ultrahigh uniform plasticity. In addition, our findings provide robust proof and atomistic visualization to verify how chemical heterogeneities regulate the deformation mechanisms and influence mechanical performance, highlighting the inherent plasticity of HEA alloys.”

Responses to Reviewer #2:

Comment #1. This paper utilizes nanoscale experimental testing inside a TEM complemented with large-scale MD simulations to conduct room temperature mechanical tensile testing of high entropy alloy nanopillars. They demonstrate that this material geometry could exhibit ultrahigh uniform tensile ductility through deformation twinning and dislocation slip. This is a noteworthy finding, and can significantly impact the design and fabrication of microelectronic devices. To the reviewer's knowledge, this work is new and is a significant advancement of prior mechanistic understanding of both high entropy alloys as well as nanostructured and nanoscale metals. Both the computational and experimental aspects are fairly complete, and conclusive.

Response: We are very grateful to you for your time and effort to review our paper and for your positive comments on our manuscript.

Comment #2. Some (optional) minor suggestions are as follows:

(1) There appears to be very good matching between the experimental observations and the MD simulation results. Can the authors confirm if changes in the atomistic configurations (from that in Fig. 3) will influence the dislocation density and twin fractions? Perhaps these can be expressed as statistical quantities with error bars (akin to Fig. 4)

Response: Thank you very much for your insightful comments. We fully agree with you that atomic configurations indeed have a certain influence on the evolution of dislocation density and twin fraction during plastic deformation. In our current study, we changed the atomic configurations of simulated samples by adjusting two parameters (including the width-thickness ratio and initial dislocation density), as shown in Fig. 3a. In response to your comments, we have extracted the evolution of dislocation density and twin fraction from our MD simulations. Figures R1a and R1b present the variations of the dislocation density and twin fraction with the tensile strain, respectively. These results have been shown in Fig. 3b and 3c. For the simulated samples with pre-existing dislocations, dislocation nucleation, reaction and annihilation lead to a certain fluctuation of dislocation density during deformation. But a few

nanoscale twins from and gradually grow up due to the successive slip of initial partial dislocations, leading to a continuous increasing of twin fraction. For the simulated samples without initial dislocations, a number of dislocations nucleate from free surface during yielding and then slip into grain interior, followed by nucleation and extension of nanoscale twins (Extended Data Fig. 7d). The nucleation of dislocations and nanoscale twins induce the sharp increase in dislocation density and twin fraction, as indicated by brown arrows in Fig. R1, respectively. After yielding, the dislocation density increases gradually and then undergoes a certain fluctuation with dislocation multiplication, reaction and annihilation, while the twin fraction keeps a continuous increase due to the extensive twinning activities (including nucleation, interaction, extension and coalescence of nanoscale twins). After yielding, all the simulated samples exhibit the similar trend in the evolution of twin fraction, but different trend in the evolution of dislocation density. These results indicate that both initial dislocation density and width-thickness ratio indeed have a significant influence on the evolution of dislocation density, but have a weak influence on the evolution of twin fraction. However, it is very challenging to quantify the profound influences of atomic configurations on the evolution of dislocation density and twin fraction, because it requires a substantial amount of simulated samples and each sample contains millions of atoms.

In Fig. 4, the generalized planar fault energy (GPFE) curve was calculated by averaging the results from 5,000 independent samples with different random solid solutions, which could cover the wide distribution of energy pathways for deformation of CoCrFeNi. Figures 3b and 3c show the variations in the evolution of dislocation density and twin fraction in the simulated samples with different atomic configurations, respectively. However, the limited number of simulated samples could not allow us to explicitly express the dislocation density and twin fraction as statistical quantities with error bars. In future, we will further investigate the influences of initial dislocation density, width-thickness ratio and other configuration parameters (including sample shape/morphology and atomic ordering) on the evolutions of dislocation density and twin fraction.

In response to your comments, we have added the following statements on page 8 in the revised manuscript to address the influence of atomic configurations on the evolutions of dislocation density and twin fraction,

“It is noted that after yielding, all the simulated samples exhibit the similar trend in the evolution of twin fraction (Fig. 3b), but distinct trend in the evolution of dislocation density (Fig. 3c). These results indicate that both initial dislocation density and width-thickness ratio have a significant influence on the evolution of dislocation density, but have a weak influence on the evolution of twin fraction. In future, we will further quantify the influences of initial dislocation density, width-thickness ratio and other configuration parameters (including sample shape/morphology and atomic ordering) on the evolutions of dislocation density and twin fraction, which is crucial for understanding the plastic deformation behaviors and mechanisms of HEAs.”

Fig. R1 | Evolution of dislocation density and twin fraction with tensile strain in the simulated nanopillars with different atomic configurations. a, Evolution of the dislocation density. **b,** Evolution of the twin fraction.

Comment #3. (2) One also does wonder how strongly the activated mechanisms depend on the alloying content. Perhaps a small set of additional simulations on regular metal alloys could be performed to demonstrate more strikingly that these mechanisms are only activated because of the presence of these multiple elements.

Response: Thank you very much for your constructive and helpful suggestions. In our original manuscript, we have investigated the tensile behaviors of single crystalline $\langle 110 \rangle$ -oriented CoCrFeNi, Ni, Al_{0.1}CoCrFeNi, and CoCrNi and further compared their differences via MD simulations, in order to show the influence of alloying content on deformation behaviors/mechanisms and elongation. Following your suggestion, we

have performed the additional atomistic simulations for the uniaxial tension in single crystalline $\langle 110 \rangle$ -oriented Al and Ag nanopillars. Al has a high stacking fault energy (SFE), while Ag has a low SFE close to CoCrFeNi. Figure R2 summarizes the results from MD simulations for uniaxial tension of $\langle 110 \rangle$ -oriented CoCrFeNi, Ni, Al_{0.1}CoCrFeNi, CoCrNi, Al and Ag nanopillars. The stress-strain curves in Fig. R2a indicate that the alloys with multiple elements exhibit a superior combination of ultrahigh flow stress and exceptional elongation. Figure R2b demonstrates that both dislocation slip and deformation twinning are activated for pure metallic nanopillars. Since Ag has the lower stacking fault energy (SFE) than Al and Ni, more dislocations and nanoscale twins are nucleated in Ag nanowires. Especially, more significant deformation twinning is activated in Ag nanowires, leading to the higher twin fraction and elongation, compared with Al and Ni nanowires. However, it is noted that in all the pure metallic nanopillars, twin propagation was severely hindered by twin-twin and twin-dislocation interactions, and then dislocation slip dominated the deformation, leading to the strain localization and final premature failure. For the HEA nanopillars, the introduction of multiple elements leads to atomic level heterogeneity and wide variations in generalized stacking fault energy and associated dislocation activities. It gives rise to the spatial and synergistic coordination of enhanced dislocation activities and deformation twinning, which effectively promote deformation delocalization and delay necking failure. Therefore, the HEA nanopillars achieve a high twin fraction and a uniform super-elongation. In addition, as exhibited in Fig. R2b and Extended Data Fig. 12, deformation twinning accompanied by FCC-HCP transformation took place in CoCrNi MEA due to its extremely low SFE (Extended Data Fig. 13), resulting in a moderate twin fraction less than 0.6 and a high HCP fraction about 0.27. Intriguingly, such slightly different plastic behavior in CoCrNi MEA can still promote strain delocalization, enabling the specimen to maintain an excellent elongation. Furthermore, the dislocation densities of MEA and HEA nanopillars were much higher than those of pure metallic nanopillars, suggesting a higher dislocation multiplication rate in the MEA and HEAs due to their compositional heterogeneities at the atomic level. Therefore, these atomistic simulations indicate the significance of local compositional

complexities in the joint and/or sequential activations of the twinning and dislocation-mediated plasticity, which stimulate strain delocalization and suppress plastic instability, endowing them with ultrahigh uniform tensile elongation and high strength.

Fig. R2 | Comparison of tensile behaviors between single crystalline $\langle 110 \rangle$ -oriented pure metals (Al, Ni and Ag), MEA (CoCrNi), and HEAs (CoCrFeNi and $\text{Al}_{0.1}\text{CoCrFeNi}$) nanopillars. a, Simulated tensile stress-strain curves. b, Simulated evolution of twin fraction with the tensile strain. The inset shows the evolution of HCP fraction with tensile strain. c, Simulated evolution of dislocation density with the tensile strain. d-f, A sequence of snapshots of the Al, Ni, and Ag nanopillars without initial dislocation during tension. g-i, A sequence of snapshots of the CoCrNi, CoCrFeNi, and $\text{Al}_{0.1}\text{CoCrFeNi}$ nanopillars without initial dislocation during tension. The twin-twin interactions and dislocation slip in d-i are indicated by yellow and black arrows, respectively.

In response to your comments, we have added the results (including stress-strain curves, the evolution of dislocation density and twin fraction, and typical snapshots of deformed samples) from MD simulations for tension of Al and Ag nanowires into Extended Data Figs. 7, 11, and 12 for comparison in the revised Supplementary

Information. The corresponding descriptions in the manuscript and figure captions in the Supplementary Information have been also updated. The updated Extended Data Figs. 7, 11 and 12 are shown below. We have also added the following statements on pages 9-10 in the revised manuscript to address the influence of presence of multiple elements on plastic deformation mechanisms,

“Al and Ni are the constitutive elements of $\text{Al}_x\text{CoCrFeNi}$ and have the higher SFEs than CoCrFeNi , while Ag has a low SFE close to that of CoCrFeNi .”

“These results also suggested that the activation of these unique deformation mechanisms is associated with the introduction of multiple elements.”

Extended Data Figure 7 | Comparison of tensile behaviors between single crystalline $\langle 110 \rangle$ -oriented CoCrFeNi HEA and pure metallic (Al, Ni, and Ag) nanopillars. a, Simulated tensile stress-strain curves of the HEA and pure metallic nanopillars. **b, c**, Simulated evolutions of twin fraction and dislocation density with the tensile strain in HEA and pure metallic nanopillars, respectively. **d**, A sequence of snapshots of the HEA nanopillar without initial dislocation during tension, indicating that plastic deformation is dominated by deformation twinning, leading to a large

uniform elongation of the HEA nanopillar at room temperature. **e**, A sequence of snapshots of the stretched Ni nanopillar, suggesting that although plastic deformation is dominated by both dislocation slip and deformation twinning, dislocation slip induces deformation localization, resulting in early failure of the Ni nanopillar at a tensile strain of $\sim 30\%$. The dislocation slip and deformation twinning in **d** and **e** are indicated by black and orange arrows, respectively. The twin-twin interactions are indicated by the yellow arrows in **d** and **e**. **f, g**, A sequence of snapshots of the stretched Al and Ag nanopillars. It indicates the occurrence of deformation localization induced by twin-twin interaction. Due to the low SFE of Ag, relatively significant dislocation slip and twinning are activated, resulting in the moderate twin fraction and elongation of Ag nanopillars.

Extended Data Figure 11 | Atomistic simulations of $\langle 110 \rangle$ -oriented $\text{Al}_{0.1}\text{CoCrFeNi}$ HEA nanopillars under uniaxial tension. **a, Tensile stress-strain curves of simulated HEA nanopillars. **b, c**, Simulated evolutions of twin fraction and dislocation density with the tensile strain in HEA nanopillars, respectively. The stress-strain curve and evolutions of twin fraction and dislocation density of **simulated nanopillars for pure metals (Al, Ni and Ag)** are included in **a-c** for comparison. **d**, A sequence of snapshots of a stretched nanopillar with $\lambda=2:1$ and $\rho_0 \neq 0$. The atoms are colored according to their von Mises shear strains and local crystalline structures, respectively. The orange arrows and black arrows indicate the deformation twinning and dislocation slip, respectively. The yellow arrows indicate the twin-twin interaction.”**

Extended Data Figure 12 | Atomistic simulations of $\langle 110 \rangle$ -oriented CoCrNi medium entropy alloy (MEA) nanopillars under uniaxial tension. a, Tensile stress-strain curves of simulated MEA nanopillars. **b, c,** Simulated evolutions of twin fraction and dislocation density with the tensile strain in MEA nanopillars, respectively. **The inset in b indicates the fraction of atoms with hexagonal close-packed (HCP) structure.** The stress-strain curve and evolutions of twin fraction, **HCP fraction** and dislocation density of **simulated nanopillars for pure metals (Al, Ni and Ag)** are included in **a-c** for comparison. **d,** A sequence of snapshots of a stretched nanopillar with $\lambda=2:1$ and $\rho_0 \neq 0$. The atoms are colored according to their von Mises shear strains and local crystalline structures, respectively. The orange arrows and black arrows indicate the deformation twinning and dislocation slip, respectively. The yellow arrows indicate the twin-twin interaction.”

Comment #4. (3) The manuscript is very well written. Couple of typos found under the methods section on "atomistic simulations" (reference source not found, and also "EAM potentia" should be "EAM potential").

Response: Thank you very much for your pointing out these issues. We have carefully checked our manuscript, updated the reference sources and fixed all the typos in the revised manuscript.

REVIEWERS' COMMENTS

Reviewer #1 (Remarks to the Author):

The authors have addressed my comments satisfactorily and I recommend the manuscript for publication.

Reviewer #2 (Remarks to the Author):

The revised manuscript is much improved and all comments from both reviewers have been addressed. Recommend publish as is.